# Feasibility of Hydrostatic Transmission in Community Wind Turbines

Yingkun Sheng, Daniel Escobar-Naranjo * and Kim A. Stelson *

Department of Mechanical Engineering, University of Minnesota, Minneapolis, MN 55455, USA; sheng087@umn.edu
* Correspondence: escob060@umn.edu (D.E.-N.); kstelson@umn.edu (K.A.S.)

**Abstract:** This study investigates the potential improvement of a community wind turbine through replacing the conventional drivetrain with a hydrostatic transmission (HST). Conventional wind turbines use a fixed-ratio gearbox, a variable-speed induction generator, and power electronics to match the grid frequency. Because of unsteady wind, the reliability of the gearbox has been a major issue. An HST, a continuously variable transmission with a high power density, can replace a conventional transmission. The resulting wind turbine has the potential to offer the advantages of a lower cost, decreased weight, and increased reliability. For the application considered in this study, the main source of LCOE increase is due to the inefficiencies in the system. Even if the cost of the proposed HST transmission is free, because of inefficiency, the levelized cost of electricity will be higher than for a turbine with a conventional fixed-ratio gearbox. For the HST solution to be cost-competitive, increases in efficiency and reductions in cost are required.

**Keywords:** wind turbine; renewable energy; hydrostatic transmission; levelized cost of electricity

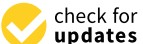



## 1. Introduction

Wind energy is the fastest-growing renewable energy source. Among all the renewable sources of electricity, wind power is the cheapest [1]. The Department of Energy has the goal of generating 20% of the United States electricity using wind power by 2030. Distributed wind turbines will play an important role in reaching this goal [2].

Wind turbines require a speed-up transmission to connect the low-speed turbine to the high-speed generator. A conventional drivetrain circuit is shown in Figure 1. Conventional wind turbines use a fixed-ratio gearbox, a variable-speed induction generator, and power electronics to match the grid frequency. The gearbox is heavy, and the unreliability of the gearbox and power electronics have caused high maintenance costs due to unsteady wind.

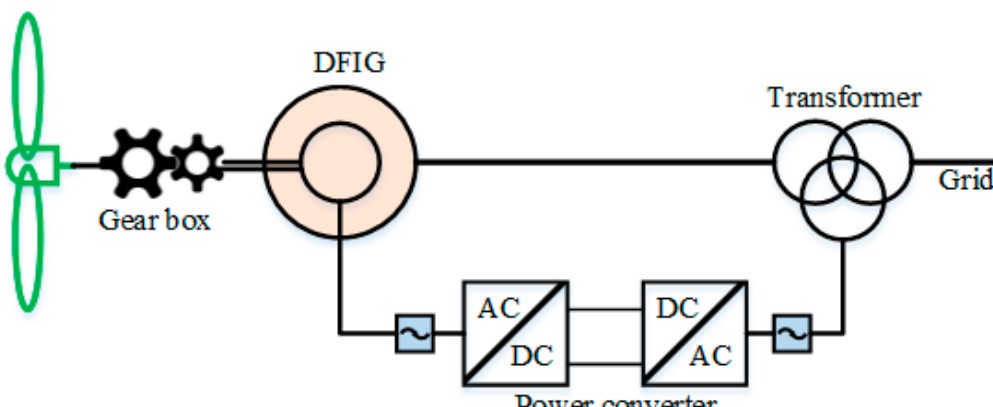

**Figure 1.** Conventional wind turbine drivetrain.

A continuously variable transmission (CVT) can replace a conventional transmission. Although mechanical CVTs are available, the high power needed for wind turbines requires hydrostatic transmissions (HSTs) [3]. A hydrostatic transmission consists of a hydraulic pump driving a hydraulic motor. To be a continuous variable transmission, one of the units must have variable displacement, where the most suitable configuration uses a large fixed-displacement pump and a small variable-displacement motor. A hydraulic circuit of an HST wind turbine is shown in Figure 2, consisting of a fixed displacement pump and variable displacement motor. The rotor turns the pump, causing fluid to flow from the pump to the motor, turning the generator. A synchronous generator replaces the conventional induction generator to better match the grid frequency, and power electronics are eliminated because the generator rotates synchronously to match the grid frequency, reducing the cost and improving the efficiency and reliability. The slight compressibility of hydraulic fluid shields the mechanical components from shock loading, improving reliability. The availability of off-the-shelf components in the community wind power range makes it a feasible option. The HST wind turbine is a simpler and more reliable system, as seen in Figure 2, but it is less efficient. The most important losses in an HST wind turbine are the viscous and leakage losses in the hydraulic pumps and motors.

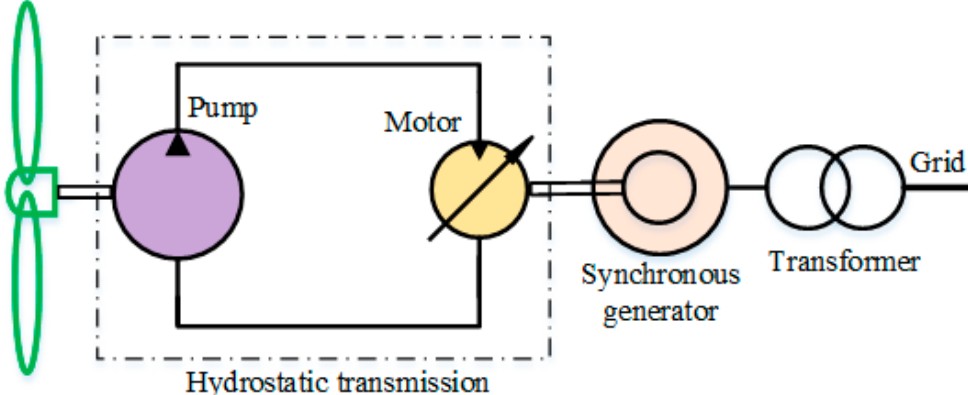

**Figure 2.** HST wind turbine.

Hydrostatic wind turbines have been extensively researched, but not commercialized. Chen et al. provide a detailed review of the research and development in this area [4]. A few examples are mentioned here. Wind turbines can be classified as large, meaning larger than 1 MW, suitable for utility use, or small, meaning smaller than 1 MW, which is suitable for community or residential use. Chapdrive is an early demonstration of a utility-scale hydrostatic wind turbine in Norway [5]. Rampen et al. [6] describe the development of the efficient digital displacement technology that was commercialized by Artemis. The Artemis technology was demonstrated by Mitsubishi in seven MW offshore wind turbines in Japan and Scotland, in a project named Sea Angel [7]. Dumnov et al. [8] demonstrated experimentally validated a digital displacement pump and showed efficiencies above 90% for multiple operating conditions. A one MW hydrostatic test stand for wind power was developed at RWTH Aachen [9]. The test stand demonstrated the possibility of switching multiple pumps and motors depending on the power level. An example of an experimental demonstration of an HST for community wind turbines is the regenerative test stand at the University of Minnesota [10]. The inclusion of an accumulator to rapidly store and reuse energy has also been studied [11,12], but it requires a large accumulator [13,14] that could significantly increase the LCOE.

Cost is a major consideration in the choice of wind turbine transmission. For utility-scale turbines, large custom components must be acquired. For community wind turbines, widely available off-the-shelf components can be used. In either application, being priced competitively with alternatives is essential.

A conventional wind turbine is manufactured by Pecos Wind Power, located in Somerville, MA, USA (see https://pecoswindpower.com (accessed on 7 November 2023)). The Pecos product, the PW85, is an 85-kW community-scale wind turbine designed to have a higher typical utilization factor, enabling it to operate economically on sites with lower average wind speeds. Unlike most commercial wind turbines, the PW85 is optimized for low wind speeds, which can reduce the cost of electricity and provide energy over more varied sites. The PW85 uses a conventional three-stage gearbox with a 56:1 gear ratio rated at 250 kW, coupled with an induction generator. Since the power level of the drivetrain is derated, the lifetime of the PW85 can be extended to twenty years [15].

An HST was designed at the University of Minnesota to fit into the PW85. Two options were investigated: an HST with a large pump and an HST with a speed-up gearbox and a smaller pump replacing the large pump. A detailed comparison is described in this study, and the feasibility is based on the efficiency, weight, capital cost, and levelized cost of electricity (LCOE). Although this paper is not an experimental paper, the analysis is based on an experimentally validated simulation model [10].

## 2. Preliminary Design of HST

### 2.1. Design One—Hydrostatic Transmission (HST)

This design consists of a main pump, charge pump, motor, and synchronous generator. The duty cycle, which is the load information for the PW85, is provided by Pecos. The duty cycle consists of the torque and speed of both the low-speed and high-speed shafts (LSSs and HSSs) for each wind speed. For an ideal pump [16], the mechanical power is entirely converted to fluid power.

$$P = T\omega = pQ \tag{1}$$

$P$ is the power [Nm s$^{-1}$], $T$ is the torque [Nm], $\omega$ is the rotation rate [rad s$^{-1}$], $p$ is the differential pressure [Nm$^{-2}$], and $Q$ is the flow rate [m$^3$ s$^{-1}$]. The displacement of the pump, $D$ [m$^3$ rev$^{-1}$], and its speed of rotation, $\omega$, determine the flow rate, $Q$.

$$Q = \frac{D\omega}{2\pi} \tag{2}$$

Combining Equations (1) and (2), we find the relationship between the pressure and the torque.

$$p = \frac{2\pi \times T}{D} \tag{3}$$

Rearranging Equation (3) to solve for the required pump displacement, $D_p$, we see that

$$D_p = \frac{2\pi \times T_r}{p} \tag{4}$$

Assuming a maximum differential pressure of 350 bar, the most common operating pressure for off-the-shelf hydraulic components, and a maximum rotor torque of 27.5 kNm, which is the maximum operating torque of the current Pecos design, the value of $D_p$ can be found:

$$D_p = 4900 \, \frac{cc}{rev} \tag{5}$$

Since the flow rate exiting the pump equals the flow rate entering the motor,

$$Q = \omega_r \times D_p = \omega_g \times x \times D_m \tag{6}$$

where $\omega_r$ is the rotational speed of the rotor, $\omega_g$ is the rotational speed of the generator, $x$ is the fractional displacement of the motor, and $D_m$ is the maximum displacement of the motor.

With both the rotational speeds provided, $D_m$ can be obtained at maximum $\omega_r$ when $x = 1$:

$$D_m = 87 \, \frac{cc}{rev} \tag{7}$$

To make up for the fluid losses through the hydraulic components, a charge pump is added, assuming that the volumetric efficiency of the HST is 96%,

$$D_{pc} = \frac{Q \times 2\%}{\omega_r} = 196 \; \frac{cc}{rev} \tag{8}$$

where $Q$ is the maximum flow rate when $\omega_r$ is at its maximum.

Additionally, the hydraulic system consists of pipelines and fittings, hydraulic oil, and a reservoir. To fit the HST into the nacelle of the turbine, a 1.8 m pipeline is needed. Using the rule of thumb that the size of the reservoir in liters should be three times the maximum flow rate in liters per minute, the reservoir size was determined to be at least 160 L.

### 2.2. Design Two—Hydrostatic Transmission with Gearbox (HST + GB)

A large hydraulic pump is expensive due to its size and low production numbers. To avoid using a large pump, a smaller pump coupled with a gearbox is considered. The gearbox reduces the displacement of the pump by a factor of the gear ratio. At the University of Minnesota, a 30:1 gearbox is available, and tests have been performed on it. Such a gearbox reduces the pump by a factor of thirty, resulting in a 166 cc/rev pump. All other components remain the same. A detailed comparison of the benefits of this design will be presented.

### 3. Assumption Validation

A comparison can be conducted based on cost analysis, which is carried out through assuming static models and fixed efficiency. To validate the assumptions of our cost analysis, we will test each assumption on the detailed numerical model described in [10]. Since detailed data on the efficiency variation of the components in the proposed Pecos design are lacking, we will perform the validating simulation on the hydrostatic wind transmission in the Hydraulic Power Transmission Lab at the University of Minnesota, where detailed manufacturer data analysis and experimental validation have already been conducted. The dynamic modeling and control of the test stand are well developed based on experiments [17]. This test stand consists of a 2512 cc/rev radial piston pump (maximum pressure 350 bar), a 135 cc/rev variable displacement axial piston motor, and a synchronous generator running at 1800 rpm [18]. Since the Pecos design is very similar in all respects, the simplified assumptions should also apply.

To validate a static model, a comparison of thirty minutes of power output using turbulent and constant wind input is shown in Figure 3. The turbulent wind profile is created using FAST code from NREL [19]. In Figure 3, to compare the steady-state power output, the transient data in the first 300 s are neglected.

To compare the energy generated between constant and turbulent wind, the energy ratio is shown in Table 1.

**Table 1.** Energy output ratio between constant and turbulent wind.

| Mean Wind Speed | 6 m/s | 7 m/s | 8 m/s | 9 m/s | 10 m/s | 11 m/s |
|---|---|---|---|---|---|---|
| Constant Wind Energy / Turbulent Wind Energy | 0.94 | 0.96 | 0.97 | 0.99 | 1.07 | 1.01 |

As Table 1 shows, the power output from constant wind is always close to that of turbulent wind with the same mean wind speed. From the plots in Figure 3, the reason for the variations in Table 1 can be understood. When the mean wind speed is below 10 m/s, the constant wind assumption underestimates the energy output, as shown by the ratio being less than one in Table 1. Because wind power varies as the cube of the wind speed, the increase in power when the wind speed is above the mean speed is greater than the decrease in power when the wind speed is below the mean speed. When the mean wind speed is 10 m/s, the power will decrease when the wind is below 10 m/s but will not

increase when the wind is above 10 m/s because the power is limited by the pitch-control system. This explains why the ratio is above one for 10 m/s. When the mean wind speed is 11 m/s, both constant and turbulent wind produce the rated power most of the time, but occasionally the turbulent wind produces less than the rated power, leading to a ratio slightly greater than one but smaller than the ratio at 10 m/s. The data in Table 1 show that the assumption of using a static model with constant wind input is appropriate.

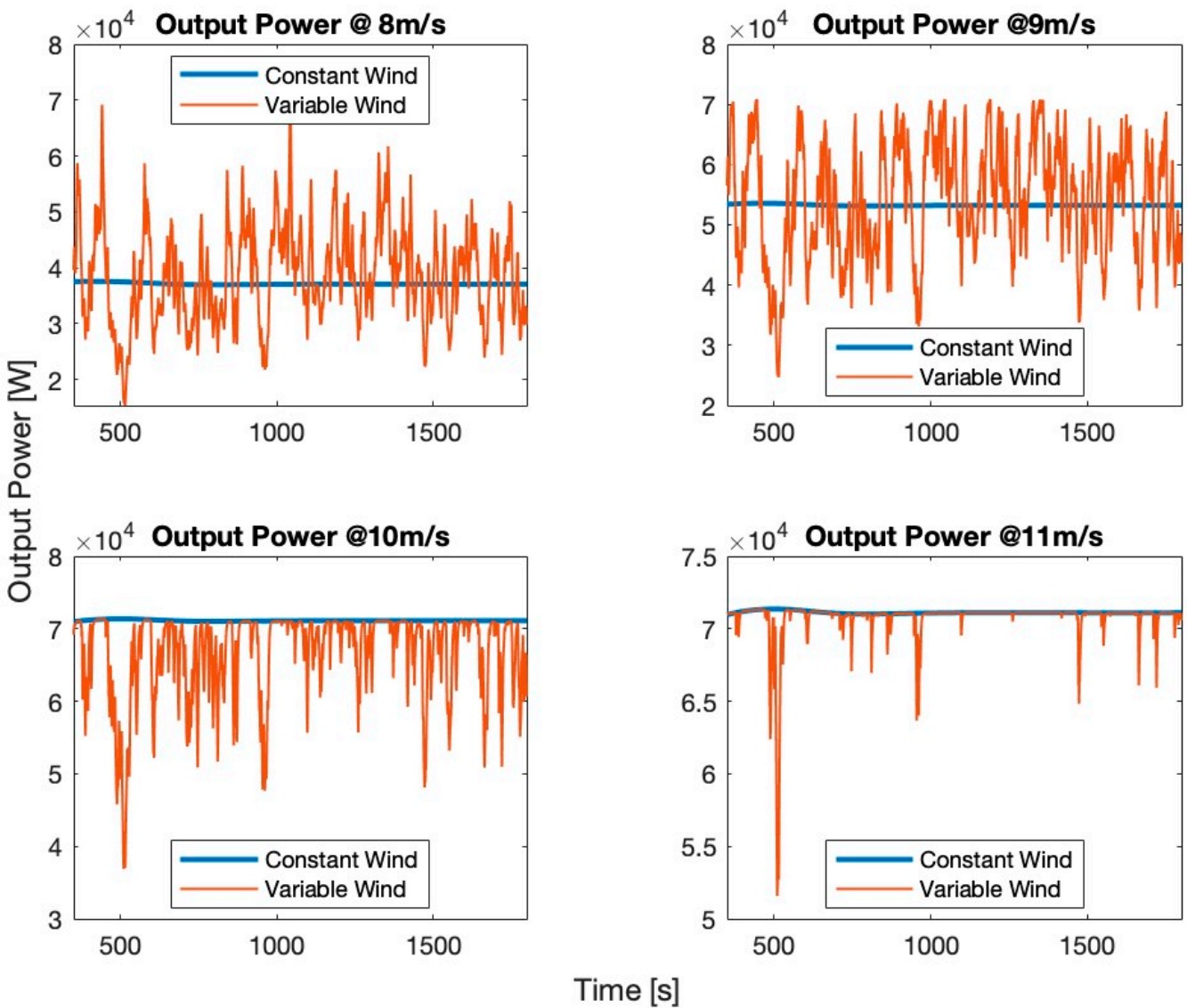

**Figure 3.** The power output of variable constant and turbulent wind inputs.

To validate that using a fixed efficiency is appropriate, a comparison of thirty minutes of power output using constant and variable efficiencies is shown in Figure 4. The inefficiencies are in the hydraulic pump and motor and are due to viscous friction and leakage. Again, the first 300 s are neglected to avoid transient effects.

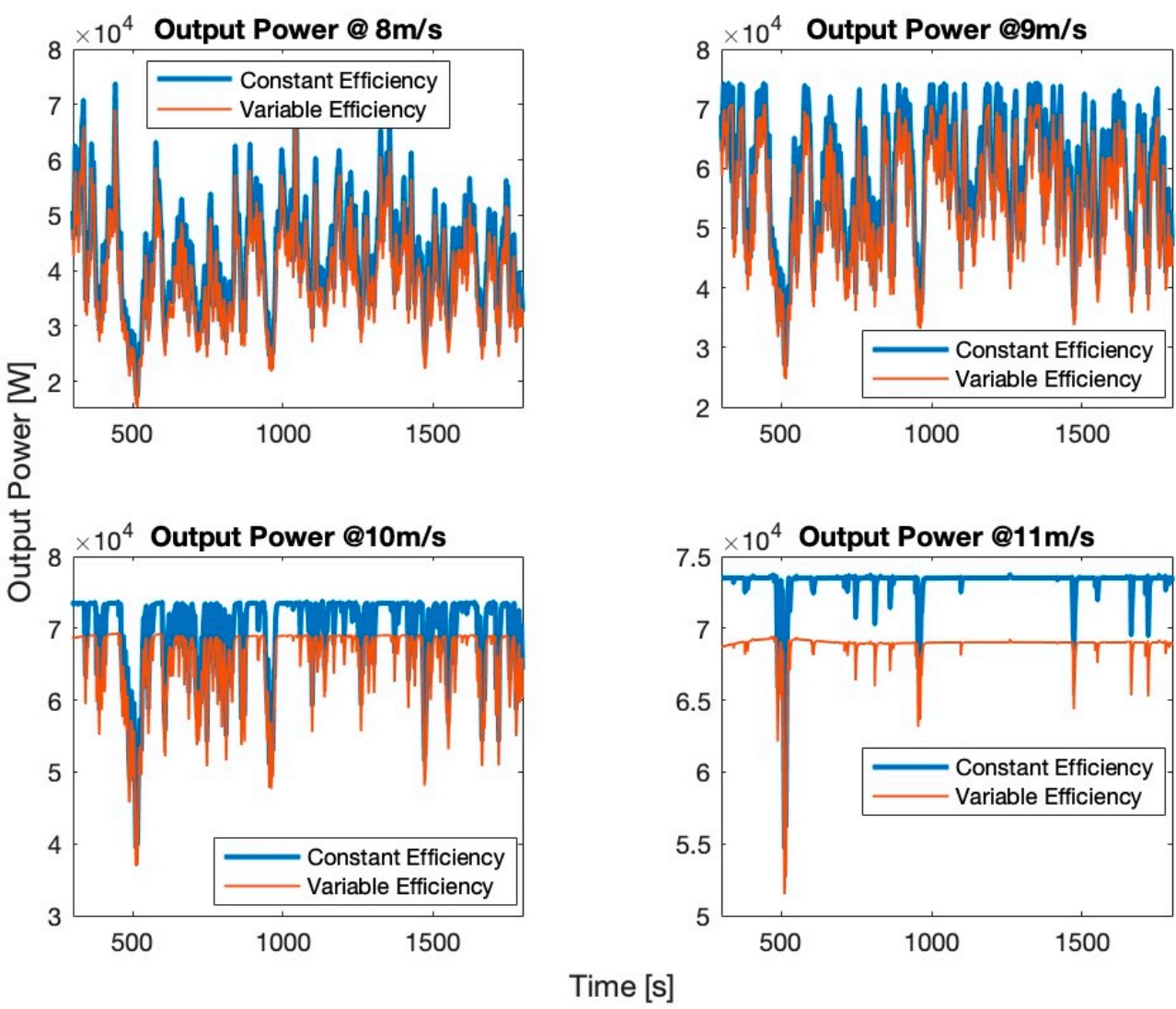

**Figure 4.** The power output of the constant and variable efficiency models.

The energy output ratio between the constant and variable efficiencies at each wind speed is shown in Table 2.

**Table 2.** Energy output comparison between constant and variable efficiencies.

| Mean Wind Speed | 6 m/s | 7 m/s | 8 m/s | 9 m/s | 10 m/s | 11 m/s |
|---|---|---|---|---|---|---|
| Constant Wind Energy / Turbulent Wind Energy | 1.15 | 1.11 | 1.08 | 1.07 | 1.07 | 1.06 |

As Table 2 shows, a model with constant efficiency would give similar results to one with variable efficiency. The constant efficiency assumption overestimates the energy output, as expected. But this assumption will be compensated, to some degree, by the underestimates from assuming constant wind. For cost analysis purposes, the simplifying assumptions are reasonable, since the uncertainties in the cost data are greater than the uncertainties in the energy output.

To understand why assuming constant peak efficiency is reasonable, consider the relationship between efficiency and wind speed, as shown in Table 3.

**Table 3.** Efficiency and motor displacement at each wind speed.

| Wind Speed | 5 m/s | 6 m/s | 7 m/s | 8 m/s | 9 m/s | 10 m/s |
|---|---|---|---|---|---|---|
| Motor Displacement | 39% | 49% | 57% | 66% | 74% | 80% |
| Efficiency | 68% | 77% | 81% | 82% | 83% | 83% |

Table 3 shows that higher wind speeds require larger motor displacements, resulting in a higher efficiency. Since wind power varies as the cube of wind speed, most of the energy comes from higher wind speeds, where the motor is highly efficient. Therefore, assuming constant peak efficiency in the cost analysis is appropriate. Hydraulic components can achieve higher efficiencies by reducing friction and leakage. New digitized hydraulic pumps and motors have the potential to overcome these problems [6,8], with efficiencies greater than 90%. These pumps and motors with the appropriate displacements for wind turbines are expected to enter the market in the near future.

## 4. Comparisons

Based on the information provided by Pecos and quotes obtained for commercial hydraulic products, the cost, peak efficiency, weight, and normalized cost of each design are listed in Tables 4–6.

Referring to Tables 4–6, the HST design has the highest overall cost due to the high price of the large pump. It has a lower efficiency than the conventional design but has a much lower weight. The HST + GB design can bring the cost down significantly but sacrifices efficiency. The normalized costs [USD/kg] of hydraulic components are much higher than those of the gearbox, rendering the costs of the HST designs less competitive.

The total cost of a community wind turbine, including the entire system, can also be estimated. Additional components include the tower, rotor, and nacelle. There are also additional costs for development, electrical infrastructure, assembly, and installation [20]. Since the additional components are shared among all designs, assuming all other costs are the same regardless of drivetrain, a total cost comparison can be estimated, as shown in Table 7.

The levelized cost of electricity (LCOE) evaluates the life cycle cost of a power plant and is primarily used for comparison with other power sources. The LCOE balances costs with energy production; thus, it considers the capital cost, finance, annual maintenance costs, and annual energy production. Based on the capital cost, a detailed levelized cost of electricity analysis can be performed using a standard approach, as shown in Equation (9) [21]. Assuming constant power output and constant operation and maintenance cost [22],

$$LCOE = \frac{FCR \times CapEx + O\&M}{NCF \times 8760} \tag{9}$$

where the *LCOE* is the levelized cost of electricity [USD/kWh]; the *FCR* is the fixed charge rate that annualizes the upfront project capital cost and accounts for return on debt and equity, taxes, and the expected financial life of the project. Assuming the *FCR* = 6% [22]; *CapEx* is the turbine capital expenditures [USD/kW]; *O&M* is the annualized operation and maintenance cost [USD/kW-yr]; and the *NCF* is the net capacity factor (scaled by 8760 h in a year), the ratio between the net turbine output energy and the turbine maximum energy that could have been generated during the same period [23].

**Table 4.** Cost, weight, and efficiency of Pecos drivetrain.

| Conventional Drivetrain | Cost [USD/Unit] | Weight [kg] | Normalized [USD/kg] | Peak Efficiency $\eta = P_{out}/P_{in}$ |
|---|---|---|---|---|
| Gearbox (56:1) | 15,750 | 1705 | 9.2 | |
| Induction generator | 7326 | 725 | 10 | |
| Power Electronics [22] | 7763 | 108 | 71.8 | |
| Total | 30,839 | 2430 | - | 92% |

**Table 5.** Cost, weight, and efficiency of HST design.

| HST | Cost [USD/Unit] | Weight [kg] | Normalized [USD/kg] | Peak Efficiency |
|---|---|---|---|---|
| pumps | 24,750 + 1400 | 360 | 70.7 | 95% |
| motor | 3462.5 | 42 | 82.4 | 90% |
| 1.8 m Pipelines + fittings | 300 | 20 | 15 | - |
| 160 L reservoir + oil | 640 + 1550 | 60 + 129.75 | 11.5 | - |
| Synchronous generator (WEG) | 5976 | 430 | 13.9 | 96% |
| Total | 38,078.5 | 1033.75 | - | 82% |

**Table 6.** Cost, weight, and efficiency of HST + GB design.

| HST + GB | Cost [USD/Unit] | Weight [kg] | Normalized [USD/kg] | Peak Efficiency |
|---|---|---|---|---|
| Gearbox (31.3:1) | 8257 | 700 | 11.8 | |
| Pumps (166 cc/rev) | 5700 | 80 | 71.3 | 80% |
| Motor | 3462.5 | 42 | 82.4 | |
| 1.8 m Pipelines + fittings | 300 | 20 | 15 | - |
| 160 L reservoir + oil | 640 + 1550 | 60 + 129.75 | 11.5 | - |
| Synchronous generator (WEG) | 5976 | 430 | 13.9 | 96% |
| Total | 25,885.5 | 1461.75 | - | 76% |

**Table 7.** Comparison on the total capital cost of all the three designs.

| Drivetrain Type | Transmission Cost (USD) | Other Components Cost (USD) [20] | Balance of System [21] | Total CapEx (USD) | Overhead (USD) |
|---|---|---|---|---|---|
| GB | 30,839 | 140,000 | 119,720 | 290,560 | |
| HST | 38,078 | 140,000 | 119,720 | 297,800 | Neglected |
| HST + GB | 25,885 | 140,000 | 119,720 | 285,600 | |

Based on the NREL cost analysis of commercial wind turbines, the component level contributions are: turbine 51.8%, balance of system 36.3%, and *O&M* 11.9% [24]. Thus, knowing that turbine cost consisting of the transmission cost and other components cost, the *O&M* cost for conventional turbine can be calculated,

$$O\&M = Turbine\ Capital\ Cost \times \frac{11.9\%}{51.8\%} = (USD\ 30839 + USD\ 14000) \times \frac{11.9\%}{51.8\%} = USD\ 39246 \qquad (10)$$

Although the *O&M* cost for HST has not been verified, it is assumed to be the same with the conventional turbine as in the worst-case scenario.

Based on the Pecos information, the *NCF* of PW85 is 45%,

$$NCF = \frac{Actual\ Power\ Out}{Max\ Power\ Out} = \frac{Actual\ Power\ In \times Efficiency}{Max\ Power\ Out} \qquad (11)$$

Since the only factor that differentiates the *NCFs* of different designs is the efficiency, the *NCFs* can be calculated for the other proposed designs:

$$NCF_{HST} = \frac{NCF_{GB}}{Efficiency_{GB}} \times Efficiency_{HST} = \frac{45\%}{92\%} \times 82\% = 40\% \tag{12}$$

Similarly,

$$NCF_{HST+GB} = 37\% \tag{13}$$

Assuming that the wind turbine has a twenty-year lifespan, the gearbox has a twenty-year service time without failure, and the HST pump needs to be rebuilt every ten years, a detailed calculation of the *LCOE* is made for the three designs:

$$LCOE1 = \frac{FCR \times CapEx + O\&M}{NCF \times 8760} = \frac{6\% \times \frac{USD\ 290560}{85\ kW} + \frac{USD\ 39246}{85\ kW\ \times\ 20\ yrs}}{0.45 \times 8760\ h/yr} = 0.0579\ [USD/kWh] \tag{14}$$

$$LCOE2 = \frac{FCR \times CapEx + O\&M + Pump\ Rebuild}{NCF \times 8760} = \frac{6\% \times \frac{USD\ 297800}{85\ kW} + \frac{USD\ 39246}{85\ kW\ \times\ 20\ yrs} + \frac{USD\ 26150\ \times\ 25\%}{85\ kW\ \times\ 20\ yrs}}{0.4 \times 8760\ h/yr} = 0.0674\ [USD/kWh] \tag{15}$$

$$LCOE3 = \frac{FCR \times CapEx + O\&M + Pump\ Rebuild}{NCF \times 8760} = \frac{6\% \times \frac{USD\ 285600}{85\ kW} + \frac{USD\ 39246}{85\ kW\ \times\ 20\ yrs} + \frac{USD\ 5700\ \times\ 25\%}{85\ kW\ \times\ 20\ yrs}}{0.37 \times 8760\ h/yr} = 0.0685\ [USD/kWh] \tag{16}$$

A detailed comparison of the LCOE is shown in Table 8.

**Table 8.** Comparisons of the LCOE.

| Transmission | Capital Cost [USD] | Efficiency | Weight [kg] | Energy [MWh/20 yrs] | LCOE [USD/kWh] |
|---|---|---|---|---|---|
| Gearbox (56:1) | 290,560 | 92% | 2430 | 6566 | 0.0579 |
| HST (4900 cc/rev pump) | 297,800 | 82% | 940 | 5852 | 0.0674 |
| HST (166 cc/rev pump) + GB (30:1) | 285,600 | 76% | 1455 | 4782 | 0.0685 |

Comparing the three designs, the conventional gearbox has the advantage of higher efficiency and the lowest LCOE. HST has the advantage of the lowest weight and highest reliability, but these are not as important as efficiency and cost for community scaled turbines. HST + GB has the lowest capital expenditure, but the highest LCOE. The main reason for the lowest LCOE being found for the conventional design is its higher efficiency, with the component price also playing a role. The hydraulic pumps and motors cost around USD 80/kg, while the gearbox costs around USD 10/kg. To be competitive, the efficiency of hydraulic components must be increased, and the price per kilogram of hydraulic components must be decreased, to compete with conventional gearbox designs.

Component prices are highly dependent on volume. The cost per kilogram of hydraulic components could decrease to the same value as a gearbox if higher production rates could be realized. To compare how much this would affect the LCOE, Table 9 shows the resulting LCOE when hydraulic components cost USD 10/kg.

**Table 9.** Comparison of the LCOE, assuming mass production.

| Transmission | Cost Decrease [USD] | LCOE [USD/kWh] |
|---|---|---|
| Gearbox (56:1) | 0 | 0.0579 |
| HST (4900 cc/rev pump) | 25,592.5 | 0.0613 |
| HST (166 cc/rev pump) + GB (30:1) | 7943 | 0.0666 |

If the turbine's gearbox fails, as they do in traditional systems [25], then the downtime would have two main consequences. The first would be the cost to repair or replace the gearbox. The second would be the power lost during maintenance. These two would negatively affect the LCOE of the gearbox system.

## 5. Alternative Scenarios

Five alternative scenarios are proposed and discussed to expand on the study and understand the causes of the differences between the LCOEs of the three systems.

The first alternative scenario assumes that the gearboxes have a ten-year life span, requiring a single gearbox replacement in the twenty-year life of the turbine. This increases the LCOE for the gearbox case from 0.0579 [USD/kWh] to 0.0602 [USD/kWh], and the LCOE for the HST + GB case from 0.0685 [USD/kWh] to 0.0700 [USD/kWh]. The result is that the total cost of replacing the gearbox does not affect the LCOE to the point where the HST becomes a better solution. The LCOE for the gearbox increases by 3.97%, and the LCOE for the HST + GB increases by 5.11%.

The second alternative scenario assumes that the hydraulic components do not need rebuilding. This reduces the LCOE for the HST from 0.0674 [USD/kWh] to 0.0663 [USD/kWh], and the LCOE for the HST + GB from 0.0685 [USD/kWh] to 0.0683 [USD/kWh]. Again, the LCOE is not significantly affected.

Figure 5 shows the impact of the reduction in the cost of the hydraulic components, the third alternative scenario. Even if the hydraulic components were free, the price reduction would not affect the LCOE enough to make the HST or HST + GB alternatives cheaper than conventional transmission.

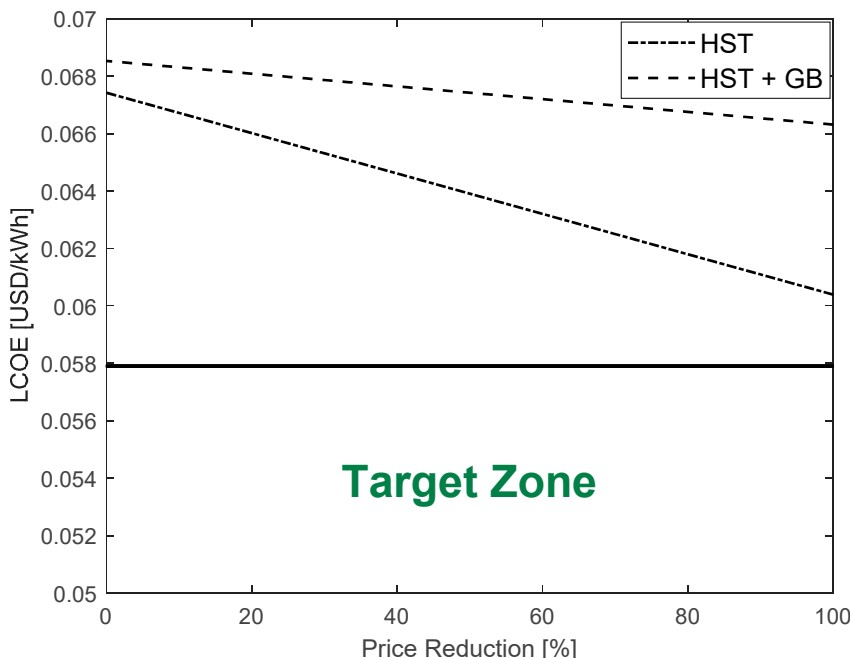

**Figure 5.** The LCOE vs. hydraulic component price reduction.

Figure 6 shows the effect of efficiency improvements to hydraulic components on the LCOE, the fourth alternative scenario. For the HST + GB system, the LCOE becomes lower than for a GB system for an efficiency greater than 91%. For the HST system, the LCOE becomes lower than that of a GB system for an efficiency greater than 95.4%.

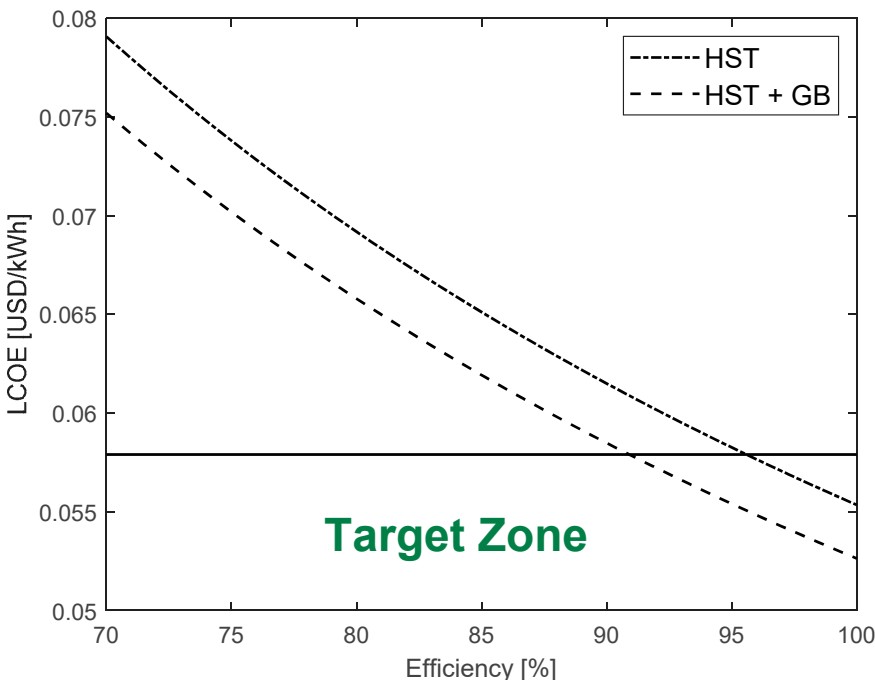

**Figure 6.** The LCOE vs. hydraulic component efficiency improvement.

The fifth alternative scenario considers changing the efficiency and the price of the hydraulic components simultaneously; see Figure 7. This scenario shows that, in practice, a combination of price reduction and efficiency improvements is needed for the HST or HST + GB solutions to have a lower LCOE than the conventional system. From Equation (9) and from Figure 7, transmission cost reduction has a much smaller effect on the LCOE than efficiency improvements.

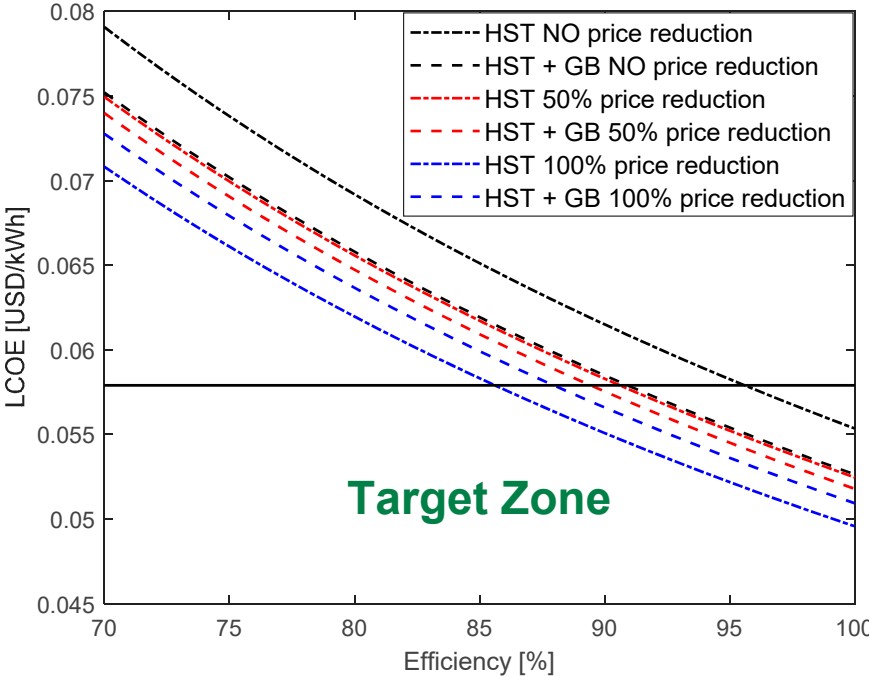

**Figure 7.** The LCOE vs. hydraulic component efficiency improvement, including price reductions for hydraulic components.

## 6. Conclusions

Our analysis finds that the current Pecos design has the lowest LCOE. The hydraulic solutions have lower efficiency and significantly higher normalized costs than the conventional drivetrain. The cost, reliability, and efficiency of hydraulic components must be improved for an HST to have a competitive LCOE for small wind turbines. List prices of hydraulic components for quantity one were used in the analysis, which means that quantity discounts could bring the cost of HST down. It has also been shown that HST can help reduce the transmission weight. Although this study focuses on replacing the current Pecos drivetrain with an HST, further research can be conducted on minimizing the cost of energy by selecting the optimal turbine rated power and rated wind speed for HST turbines [24]. It is also important to consider the possibility of downtime due to gearbox failures on conventional turbines [25]. This would increase the LCOE for conventional turbines, allowing HST turbines to be more competitive.

One of the challenges of new product introductions is that price competitiveness requires a certain minimum production quantity. The mass adoption of smaller wind turbines could dramatically reduce cost, regardless of the drivetrain. Since wind power is the fastest growing source of green electricity, large cost reductions are possible. It is unknown whether HST improvements in cost, reliability, and efficiency can make them competitive. It is also possible that because of its lighter weight, an HST wind turbine could be used in niche applications. An HST wind turbine could more easily be transported to remote areas, rendering it useful for disaster relief or military applications. If such a market could be established, then further technical improvements might be possible, enabling future expansion into new markets.

**Author Contributions:** Conceptualization, K.A.S.; Methodology, Y.S. and K.A.S.; Software, Y.S. and D.E.-N.; Validation, Y.S., D.E.-N. and K.A.S.; Formal analysis, Y.S. and D.E.-N.; Investigation, Y.S. and D.E.-N.; Resources, D.E.-N.; Data curation, Y.S. and D.E.-N.; Writing—original draft, Y.S.; Writing—review & editing, Y.S., D.E.-N. and K.A.S.; Supervision, K.A.S.; Project administration, K.A.S. All authors have read and agreed to the published version of the manuscript.

**Funding:** This research received no external funding.

**Data Availability Statement:** 3rd Party Data. Restrictions apply to the availability of these data. Data was obtained from Pecos Wind Power and are available from the authors with the permission of Pecos Wind Power.

**Conflicts of Interest:** The authors declare no conflict of interest.

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
