# Peer review of "Feasibility of Hydrostatic Transmission in Community Wind Turbines"

_actuators, doi:10.3390/act12110426_

Round 1
Reviewer 1 Report
Comments and Suggestions for Authors
Dear Authors,
The article sent for review entitled: Feasibility of Hydrostatic Transmission in Community Wind 2
Turbines presents an interesting approach to the topic presented.
The paper consists of 5 chapters, which are structured into a logical whole. The article in its current form needs a major overhaul for better readability. Attaching my comments, I encourage the authors to take them into account in order to improve its attractiveness to the reader:
- Do not end figure captions with periods
- Mathematical formulas should be centered
- Numbering of formulas should be on the right side
- The font in the formulas should be the same. Please use a different editor
- In line 126 there is a large distance between the text and the variable
- Do not end table captions with periods
- In order to better understand the issue presented, it is recommended to expand the discussion of the results
- The literature should be collated according to the orders of the journal
- Please consider increasing the literature items
- In Figures5, 6, 7, please enlarge the legend for better readability
Conclusion:
In order to continue the publication process further, the reviewer would like to encourage the authors to comply with the comments/suggestions. A thorough revision of the paper is required.
Comments on the Quality of English LanguageMinor editing of English language required
Reviewer 2 Report
Comments and Suggestions for Authors
Justify the selection of 350 bar and 27.5 kNm as design parameters for the HST.
Describe the tool employed to obtain the results in Figures 3 and 4.
Comments on the Quality of English Language
English use is sufficiently accurate.
Reviewer 3 Report
Comments and Suggestions for Authors
Present form of the paper needs improvement via mention following points:
-Introduction is not up to mark, recent related studies should be added and discussed in introduction.
-Experiment section is week, please compare the proposed framework with other methods
-What is the primary motivation behind this study in considering a hydrostatic transmission (HST) for community wind turbines, and what are the key issues associated with conventional drivetrains?
-Could you explain the fundamental differences between a fixed-ratio gearbox and an HST in the context of wind turbine applications, highlighting their respective advantages and disadvantages?
-How does the introduction of an HST impact the potential cost, weight, and reliability of a community wind turbine? What are the key factors influencing these changes?
-In the paper, it is mentioned that inefficiencies in the system represent the main source of Levelized Cost of Electricity (LCOE) increase. Could you provide more insights into these inefficiencies and their impact on LCOE?
-Can you elaborate on the conditions or assumptions under which the study suggests that the proposed HST transmission would be less cost-competitive compared to a turbine with a conventional fixed-ratio gearbox?
-What are the specific areas or components of the HST system where efficiency improvements and cost reductions are most needed to make it a cost-effective solution for community wind turbines?
Round 2
Reviewer 3 Report
Comments and Suggestions for Authors
The authors had respond to my comments